# The epidemiology of behavioral risk factors for noncommunicable disease and hypertension: A cross-sectional study from Eastern Uganda

Dustin G. Gibson[1], Ankita Meghani[1], Charles Ssemagabo[2], Adaeze Wosu[1], Gulam Muhammed Al Kibria[1]*, Tryphena Nareeba[2], Collins Gyezaho[2], Edward Galiwango[2], Judith Kaija Nanyonga[2], George W. Pariyo[1], Dan Kajungu[2], Elizeus Rutebemberwa[2], Adnan Ali Hyder[3]

1 Department of International Health, Johns Hopkins University Bloomberg School of Public Health, Baltimore, Maryland, United States of America, 2 Department of Disease Control and Environmental Health, Makerere University College of Health Sciences, Mulago Hill, Kampala, Uganda, 3 Department of Global Health, Office of the Dean, Milken Institute of Public Health, Washington, District of Columbia, United States of America

* gkibria1@outlook.com

**Data Availability Statement:** Data is available at https://www.openicpsr.org/openicpsr/workspace?

## Abstract

In light of the suboptimal noncommunicable disease (NCD) risk factor surveillance efforts, the study's main objectives were to: (i) characterize the epidemiological profile of NCD risk factors; (ii) estimate the prevalence of hypertension; and (iii) identify factors associated with hypertension in a peri-urban and rural Ugandan population. A population-based cross-sectional survey of adults was conducted at the Iganga-Mayuge Health and Demographic Surveillance System site in eastern Uganda. After describing sociodemographic characteristics, the prevalence of NCD risk factors and hypertension was reported. Prevalence ratios for NCD risk factors were calculated using weighted Poisson regression to identify factors associated with hypertension. Among 3220 surveyed respondents (mean age: 35.3 years (standard error: 0.1), 49.4% males), 4.4% were current tobacco users, 7.7% were current drinkers, 98.5% had low fruit and vegetable consumption, 26.9% were overweight, and 9.3% were obese. There was a high prevalence of hypertension and prehypertension, at 17.1% and 48.8%, respectively. Among hypertensive people, most had uncontrolled hypertension, at 97.4%. When we examined associated factors, older age (adjusted prevalence ratio (APR): 3.1, 95% CI: 2.2–4.4, APR: 5.2, 95% CI: 3.7–7.3, APR: 8.9, 95% CI: 6.4–12.5 among 30–44, 45–59, and 60+-year-old people than 18–29-year-olds), alcohol drinking (APR: 1.6, 95% CI: 1.3–2.0, ref: no), always adding salt during eating (APR: 1.6, 95% CI: 1.1–2.2, ref: no), poor physical activity (APR: 1.3, 95% CI: 1.1–1.6, ref: no), overweight (APR: 1.3, 95% CI: 1.1–1.5, ref: normal weight), and obesity (APR: 2.0, 95% CI: 1.6–2.4, ref: normal weight) had higher prevalence of hypertension than their counterparts. The high prevalence of NCD risk factors highlights the immediate need to implement and scale-up population-level strategies to increase awareness about leading NCD

goToPath=/openicpsr/200101&goToLevel=
project#.

**Funding:** Research reported in this publication was
supported by the Fogarty International Center of
the National Institutes of Health under Award
Number (DUNS Number001910777, Project #
1R21TW010415-01) and the National Science
Foundation (DG to AAH). The funders had no role
in study design, data collection and analysis,
decision to publish, or preparation of the
manuscript.

**Competing interests:** The authors have declared
that no competing interests exist.

risk factors in Uganda. These strategies should be accompanied by concomitant investment
in building health systems capacity to manage and control NCDs.

## Introduction

Each year, over 41 million people globally die from noncommunicable diseases (NCDs),
which equates to 73% of the global mortality burden and a 20% increase in the past 10 years
[1]. Approximately two-thirds of these NCD deaths are attributed to four modifiable behavioral risk factors: physical inactivity, dietary risk, alcohol consumption, and tobacco use [1–4].
These risk factors, in addition to excessive salt intake, can contribute to raised blood pressure
(BP) or hypertension, which is a major cause of all-cause deaths, estimated at 10.4 million in
2017 [3, 4].

Uganda, like many low- and middle-income countries (LMICs), is undergoing an epidemiologic transition, shifting from a primary burden of infectious diseases towards preventable
death and disability from NCDs [5–7]. NCDs are four of the ten most common causes of adult
death, and among them, cardiovascular disease is the top cause, with age-standardized death
rates of 338 per 100,000 people [8]. Raised BP, which is a leading cardiovascular risk worldwide, has also been a cause of growing public health concern in Uganda [9].

To quantify the magnitude and scope of NCD burden in Uganda, the national government,
with the support of the World Health Organization (WHO) implemented its first nation-wide
survey in 2014 to establish baseline prevalence estimates for NCD risk factors in the country as
well as to examine the behaviors, knowledge, and practices of Ugandan communities regarding NCDs [9, 10]. The results were striking: nearly one in ten Ugandans had more than three
risk factors for NCDs, which included inadequate diet, low physical activity, being overweight
or obese, or having raised BP. The survey results revealed a high burden of hypertension
(26.4%), and further analyses showed a higher prevalence observed in urban areas (28.9%)
compared to rural areas (25.8%), and a higher prevalence among men (28.3%) compared to
women (25.2%). However, among those who had hypertension, roughly 8% were aware of
their own status, thereby indicating the high burden of uncontrolled hypertension, and pointing to an immediate need to increase earlier detection of hypertension [9, 10].

These baseline survey findings were pivotal in providing the key benchmarks to track
NCD risk factors in the Ugandan population and informed the design of national-level NCD
policy guidelines to address key NCD risk factors, including the importance of creating more
public awareness about the NCDs and their risk factors [9, 10]. Additional studies have
described the barriers to hypertension prevention and management in rural settings, emphasized the need to increase the monitoring and screening of NCD risk factors at community
levels through village health teams, and underscored broader access to essential NCD medicines, particularly in rural areas [11–13]. However, there have been a limited number of
studies that examined the prevalence of hypertension and NCD risk factors together, along
with factors associated with hypertension. Therefore, studies required to examine the risk
factors in countries with increasing NCD burden, including Uganda. In light of the suboptimal NCD risk factor surveillance efforts and recognizing the need for continuous monitoring of NCD risk factors, particularly undiagnosed hypertension, the study's main objectives
were to: (i) characterize the epidemiological profile of NCD risk factors; (ii) estimate the
prevalence of hypertension; and (iii) identify factors associated with hypertension in a peri-urban and rural Ugandan population.

## Methods

### Study design, settings, and participants

This was a population-based cross-sectional survey of adults aged 18 years and older. It was nested in the longitudinal cohort of the Iganga-Mayuge Health and Demographic Surveillance System field site (IMHDSS) in eastern Uganda. Since 2004, IMHDSS staff conduct bi-annual visits to collect vital events and demographic information from a peri-urban and rural population of over 90,000 individuals residing in 17,000 households [14, 15]. The site spans sixty-five villages across seven sub-counties, and it is served by sixteen health centers and two hospitals. Each village, household, and resident are enumerated by IMHDSS staff, and these records are maintained in the IMHDSS database. Residents are assigned a unique ID that facilitates longitudinal monitoring. The database of all individuals, including their date of birth and sex, constituted the sampling frame for this study. Specifically, participants were eligible for the study if they were 18 years of age or older and had been IMHDSS residents for at least 6 months. Data collection took place between 06 November 2017 and 22 June 2018.

### Ethics statement

Ethical approval was obtained from the Makerere University School of Public Health Higher Degrees, Research and Ethics Committee, protocol (IRB: 00005876). They approved the study protocol and questionnaires. The Institutional Review Board of the Johns Hopkins Bloomberg School of Public Health exempted the study from ethical oversight and agreed to allow the study to rely on the Ethical Review Board of the Makerere University School of Public Health (IRB: 00000112 and 00000758).

### Data collection

IMHDSS staff visited individuals selected from the IMHDSS database and obtained informed written consent. Consent and all data collection occurred at the participant's residence. After consent was obtained, IMHDSS staff administered an abridged version of the World Health Organization's (WHO) STEPwise approach to NCD risk factor surveillance (STEPS) instrument, using a tablet programmed with Open Data Kit (ODK). This instrument contained questions on demographics, tobacco use, alcohol consumption, dietary intake, and physical activity [16]. Following the administration of the questionnaire and at the same household visit, trained nurses measured the participant's BP, weight, and height. We did not collect physical measurements from pregnant women.

BP was measured three times, spaced ten minutes apart, using the Omron 5 series upper arm monitor. The upper arm of the left side was used. For those who had physical measurements taken, if they were found to have high blood pressure but were in no acute danger of an event from BP (i.e., systolic BP of 140 mmHg to 179 mmHg, and a diastolic BP less than 120 mmHg), the research staff provided the participant with information on the importance of keeping optimal BP levels. However, if a participant was observed to be in hypertensive crisis (i.e., systolic or diastolic BP at least 180 or 120 mmHg, respectively), arrangements were made to transport the person to the nearest health facility if they were willing. According to Uganda's Ministry of Health guidelines, government health facilities are expected to provide free health services, including medication for hypertension [17]. Anthropometric measurements of body height and weight were assessed using standard protocols, with participants standing upright, not wearing shoes, and wearing light-weight clothes. Height was measured twice, to the nearest 0.1 centimeter (cm). Body weight was measured twice to the nearest 0.1 kilogram (kg), using

calibrated digital weighing scales. The participants were given feedback on the results of all the measurements immediately before the data collector left the household.

## Outcomes

The average of the three BP readings was used to produce the final average systolic and diastolic BP estimates. Participants were classified as hypertensive and pre-hypertensive based on WHO-International Society of Hypertension (WHO-ISH) guidelines [18, 19]. The primary outcome, hypertension, was defined as having a systolic or diastolic BP at least 140 or 90 mmHg, respectively, or if participants were currently on anti-hypertensive medication. Pre-hypertension was defined as having an systolic BP of 120 to 139 mmHg and/or diastolic BP of 80 to 89 mmHg [18, 19].

Indicator variables for NCD risk factors were created in accordance with the STEPS analysis guide [16]. Current drinkers were defined as those who drank alcohol in the past 30 days. Heavy episodic drinking was defined as those who drank six or more standard alcoholic drinks on a single occasion in the past 30 days. Low fruit and vegetable consumption was defined as those who ate less than 5 servings of fruit and/or vegetables on average per day in a week. A standard serving of fruit can be a medium-sized apple, banana, orange, or guava, or around half a cup of cooked or chopped fruit. Similarly, a typical serving of vegetables might be about a cup of leafy greens or salad, or half a cup of cooked or chopped vegetables. Given the low observed prevalence of tobacco smokers and smokeless tobacco use, we used current tobacco use (any type) in the regression models. The Global Physical Activity Questionnaire (GPAQ) analysis guide was used to create indicators for the level of physical activity [20]. Minutes spent doing physical activity were multiplied by a metabolic equivalent (MET) depending on the type of activity; 8 MET for vigorous intensity activities and 4 MET for moderate intensity activities. Insufficient physical activity, as defined by WHO, was defined as individuals who did not meet any of the following criteria: 150 minutes of moderate-intensity physical activity per week, 75 minutes of vigorous-intensity physical activity per week, or an equivalent combination of moderate- and vigorous-intensity physical activity accumulating at least 600 metabolic equivalent-minutes per week.

For anthropometric measurements, we used the average of the two measurements to produce the final standing height and body weight estimates. Body Mass Index (BMI; kilograms/meters$^2$ (kg/m$^2$)) was calculated and categorized into underweight (BMI <18.5), normal weight ($\geq$18.5 BMI < 25), overweight ($\geq$25 BMI < 30), and obese (BMI $\geq$30) in accordance with WHO recommendations [21].

## Data analysis

First, we reported the sociodemographic characteristics of the respondents as per the status of hypertension (i.e., no hypertension, prehypertension, and hypertension); we used mean (with standard error (SE)) to report continuous variables and weighted percentages (%) with unweighted numbers (n) to report categorical variables. Continuous and categorical variables were compared by analysis of variance and chi-square tests, respectively. Then, we reported the overall and sex-stratified prevalence of risk factors for NCD as well as hypertension with 95% confidence intervals (CI). Lastly, to identify factors associated with hypertension, we calculated crude and adjusted prevalence ratios (PR) for each of the NCD risk factors using weighted Poisson regression with sample selection weights. Poisson regression was used due to convergence issues with log-binomial models. Only those demographic and NCD risk factor indicators that had significant associations in unadjusted analyses (p<0.5) were included in the adjusted (i.e., multivariable) model. From the list of tobacco-related variables, we only

included those who were 'current tobacco users' as the unadjusted direction of association was similar for both smoked and smokeless tobacco use variables. Similarly, we only included one alcohol variable (i.e., current drinking) and one salt variable (i.e., add salt while eating). All statistical analyses were performed using Stata SE version 15. An alpha of 0.05 was assumed for all statistical tests of significance.

## Sample size

Using the equation for sample size (n) with stratified random sampling, 385 IMHDSS adults were needed to estimate an indicator with an expected prevalence of 50% risk factor (p = 0.5), a 5% margin-of-error ($\delta$ = 0. 05), and a 5% type-I error ($\alpha$ = 0. 05) for each of the 8 strata-male and female; age groups of 18–29 years, 30–44, 45–59, and 60+ years.$n = z_{1-\frac{\alpha}{2}}^2 \sigma^2 / \delta^2$. Following eight age-sex strata, we needed a total of 3080 (i.e., 385 * 8) participants.

## Results

Table 1 shows the sociodemographic characteristics of the respondents as per their hypertension status. A total of 3220 people were included in the analyses; the mean age of the participants was 35.3 years (SE: 0.1); the mean age (SE) of hypertensive people was higher than those with normal BP or prehypertension, 48.7 (0.7), 33.8 (0.3), and 31.7 (0.3), respectively (p<0.001). About 49.4% of the people were female. The proportion of people without any formal education was 9.1%; this proportion was higher among those with hypertension (17.5%) than people without hypertension (6.7%) or prehypertension (7.8%). A majority of the

**Table 1. Comparison of the study sample according to hypertension status.**

| Variable | | Overall | Blood pressure levels | | | p-value |
|---|---|---|---|---|---|---|
| | | | Normal | Prehypertension | Hypertension | |
| Age (in year) | Mean (SE) | 35.3 (0.1) | 31.7 (0.3) | 33.8 (0.3) | 48.7 (0.7) | <0.001 |
| | 18–29 | 46.2 (694) | 57.7 (294) | 49.4 (362) | 14 (38) | <0.001 |
| | 30–44 | 29 (727) | 26.9 (220) | 30 (371) | 30.7 (136) | |
| | 45–59 | 16.3 (823) | 12 (203) | 14.8 (359) | 29.3 (261) | |
| | 60+ | 8.5 (883) | 3.5 (121) | 5.8 (296) | 26 (466) | |
| Sex | Female | 49.4 (1626) | 58.8 (485) | 42.3 (639) | 50.8 (502) | <0.001 |
| | Male | 50.6 (1501) | 41.2 (353) | 57.7 (749) | 49.2 (399) | |
| Edu-cation | No education | 9.1 (577) | 6.7 (112) | 7.8 (220) | 17.5 (245) | <0.001 |
| | Primary | 43.6 (1480) | 41.6 (387) | 43.3 (655) | 48.5 (438) | |
| | Secondary | 38.5 (850) | 42 (275) | 40.7 (416) | 25.4 (159) | |
| | Above secondary | 8.8 (220) | 9.7 (64) | 8.2 (97) | 8.6 (59) | |
| Marital status | Never married | 28.5 (435) | 32.4 (166) | 32.4 (240) | 9.9 (29) | <0.001 |
| | Married | 60.5 (2020) | 59.9 (550) | 58.6 (901) | 67.4 (569) | |
| | Divorced/separated | 5.6 (241) | 4.8 (54) | 5.3 (104) | 8.1 (83) | |
| | Widowed | 5.3 (431) | 2.9 (68) | 3.8 (143) | 14.6 (220) | |
| Location | Rural | 58.7 (2035) | 63.9 (581) | 54.6 (839) | 60.3 (615) | <0.001 |
| | Peri-urban | 41.3 (1092) | 36.1 (257) | 45.4 (549) | 39.7 (286) | |
| Wealth index | Lowest | 19.9 (632) | 21.4 (184) | 17.4 (229) | 23.8 (219) | 0.21 |
| | Lower | 20.7 (520) | 19.3 (133) | 22.5 (251) | 18.6 (136) | |
| | Middle | 21.6 (576) | 21.7 (158) | 21.9 (262) | 20.8 (156) | |
| | Higher | 22.5 (506) | 22 (134) | 23.6 (236) | 20.3 (136) | |
| | Highest | 15.3 (321) | 15.7 (85) | 14.6 (141) | 16.5 (95) | |

**Table 2. Weighted estimates of noncommunicable disease risk factors by sex.**

| Indicator | Female | Male | Both sexes |
|---|---|---|---|
| Tobacco use | | | |
| Current tobacco smoker | 0.7 (0.4,1.1) | 7.6 (6.3,9.1) | 4.0 (3.4,4.8) |
| Daily tobacco smoker | 0.3 (0.2,0.6) | 5.0 (3.9,6.3) | 2.6 (2.1,3.2) |
| Current smokeless tobacco | 0.1 (0.1,0.3) | 1.4 (0.8,2.4) | 0.7 (0.5,1.2) |
| Daily smokeless tobacco | 0.1 (0.0,0.2) | 0.6 (0.3,1.4) | 0.3 (0.2,0.7) |
| Current tobacco use (any) | 0.7 (0.4,1.1) | 8.4 (7.0,10.0) | 4.4 (3.7,5.3) |
| Alcohol consumption | | | |
| Current drinker (past 30 days) | 4.6 (3.6,5.8) | 11.1 (9.4,13.0) | 7.7 (6.7,8.8) |
| Heavy episodic drinking* | 4.0 (3.0,5.2) | 9.4 (7.8,11.2) | 6.6 (5.7,7.7) |
| Diet | | | |
| Low fruit & vegetable consumption† | 98.1 (97.1,98.7) | 98.9 (98.1,99.3) | 98.5 (97.9,98.9) |
| Always/often add salt while eating | 4.1 (3.1,5.6) | 5.0 (3.9,6.6) | 4.6 (3.7,5.6) |
| Always/often add salt while cooking | 45.5 (42.6,48.5) | 48.1 (45.0,51.1) | 46.8 (44.7,48.9) |
| Always/often eat processed foods high in salt | 7.1 (5.7,8.9) | 13.4 (11.4,15.8) | 10.2 (8.9,11.6) |
| Physical Activity | | | |
| Insufficient physical activity‡ | 9.9 (8.4,11.7) | 5.9 (4.7,7.3) | 8.0 (7.0,9.1) |
| Body Mass Index (BMI; kg/m$^2$) § | | | |
| Underweight (BMI< 18.5) | 2.0 (1.4,2.8) | 2.3 (1.6,3.3) | 2.1 (1.6,2.7) |
| Normal (≥18.5 BMI < 25) | 52.6 (49.5,55.6) | 70.5 (67.8,73.2) | 61.7 (59.6,63.7) |
| Overweight (≥25 BMI < 30) | 30.7 (28.0,33.6) | 23.1 (20.7,25.7) | 26.9 (25.0,28.8) |
| Obese (BMI ≥30) | 14.7 (12.8,16.8) | 4.1 (3.1,5.2) | 9.3 (8.2,10.5) |

Data are prevalence (95%CI) and are weighted to 2016 IM-HDSS population by age-sex.

* Percent of population who had six or more alcoholic drinks in one sitting in the past month.

†Defined as those who ate less than 5 servings of fruit and/or vegetables on average per day.

‡Defined as not achieving 150 minutes of moderate-intensity physical activity OR 75 minutes of vigorous-intensity physical activity OR an equivalent combination of moderate- and vigorous-intensity physical activity achieving at least 600 MET-minutes.

§ Excludes pregnant women.

Abbreviations: kg, kilogram; m, meter.

respondents were married (60.5%) or were living in rural regions (58.7%); these characteristics also differed by hypertension status. The wealth index did not differ by hypertension status. We also reported the demographics by sex (S1 Table).

Table 2 presents the population prevalence of tobacco use, alcohol consumption, diet, physical activity, and BMI weighted to 2016 IMHDSS population by age and sex. Males had a higher prevalence of current tobacco smokers (7.6%, 95%CI: 6.3-9.1%), current smokeless tobacco users (1.4%, 95% CI: 0.81-2.4%), current drinkers (11.1%, 95% CI: 9.4-13.0%), and heavy episodic drinking (9.4%, 95% CI: 7.8-11.2%), than females, (0.65%, 95% CI: 0.39%-1.1%), (0.13%, 95% CI: 0.06%-0.31%), (4.6%, 95% CI: 3.6-5.8%), and (4.0%, 95% CI: 3.1%-5.2%) respectively. Females had higher prevalence of insufficient physical activity (9.9%, 95% CI: 8.4%-11.7%), and obesity (14.7%, 95% CI: 12.9%-16.8%), as compared to males, (5.9%, 95% CI: 4.7%-7.2%) and (4.1%, 95% CI: 3.1%-5.2%), respectively. Population prevalence estimates for low fruit and vegetable consumption, adding salt while eating, and adding salt while cooking were similar between males and females; the overall prevalence for these indicators, respectively, was (98.4%, 95% CI: 97.9%-98.9%), (4.6%, 95% CI: 3.7%-5.6%), and (46.8%, 95% CI: 44.7%-48.8%).

**Table 3. Weighted distribution of blood pressure levels by sex.**

| Indicator | Female | Male | Both sexes |
|---|---|---|---|
| **Blood pressure field reading** | | | |
| Normal blood pressure* | 40.5 (37.6–43.5) | 27.7 (25.0–30.6) | 34.0 (32.0–36.1) |
| Pre-hypertension† | 41.9 (38.9–44.9) | 55.7 (52.7–58.6) | 48.8 (46.8–51.0) |
| Hypertension‡ | 17.6 (15.9–19.4) | 16.6 (14.8–18.6) | 17.1 (15.9–18.4) |
| **Of those who had hypertension** | n = 502 | n = 399 | n = 901 |
| Newly identified hypertension case§ | 52.4 (47.9–56.8) | 74.9 (70.4–79.1) | 62.4 (59.1–65.5) |
| Uncontrolled hypertension¶ | 96.4 (93.8–98.0) | 98.4 (96.2–99.3) | 97.4 (95.8–98.4) |
| Controlled hypertension** | 3.6 (2.0–6.2) | 1.6 (0.69–3.8) | 2.6 (1.6–4.2) |

Data are prevalence (95%CI) and are weighted to 2016 IM-HDSS population by age-sex. Excludes pregnant women.

* SBP <120 mm Hg and DBP <80 mm Hg and not taking antihypertensive medication.

† Either SBP between 120 and 139 mmHg or DBP between 80 and 89 mmHg.

‡SBP ≥ 140 and/or DBP ≥ 90 mmHg or currently on medication for raised blood pressure.

§ SBP ≥ 140 and/or DBP ≥ 90 mmHg and not currently on medication for raised blood pressure and never been diagnosed with high blood pressure

¶SBP ≥ 140 and/or DBP ≥ 90 mmHg and not currently on medication for raised blood pressure.

** SBP< 140 and DBP <90 mmHg and currently on medication for raised blood pressure.

Overall, the weighted prevalence of hypertension was estimated to be 17.1% (95% CI: 15.9%-18.4%), and was similar between males (16.6%, 95% CI: 14.8%-18.6%) and females (17.6%, 95% CI: 15.9%-19.4%; Table 3). A higher prevalence of pre-hypertension, defined as having either systolic BP between 120 and 139 mmHg or diastolic BP between 80 and 89 mmHg was observed in males (55.7% 95% CI: 52.7%-58.7%) as compared to females (41.9% 95% CI: 38.9%-44.9%). Of the 399 male and 502 female hypertensive participants, 83.3% and 57.8% were newly identified as being hypertensive (i.e., had not received a prior hypertension diagnosis). The population prevalence of uncontrolled hypertension, i.e., being hypertensive but not on medications was high in both males (98.4% 95%CI: 96.2%-99.3%) and females (96.4% 95%CI: 93.8%-98.0%).

Table 4 presents a summary of the unadjusted and adjusted associations of potential risk factors for hypertension. Several factors were determined to be significantly associated with hypertension in adjusted models, including age, drinking alcohol in the last month, adding salt while eating, poor physical activity, and being overweight or obese. For instance, compared to 18-29-year-old people, those with 30–44 (adjusted PR (APR): 3.1, 95% CI: 2.2–4.4, p<0.001), 45–59 (APR: 5.2, 95% CI: 3.7–7.3, p<0.001), and 60+ years of age (APR: 8.9, 95% CI: 6.4–12.5, p<0.001) had a higher prevalence of hypertension. Those with current alcohol drinking (APR: 1.6, 95% CI: 1.3–2.0, p<0.001) or poor physical activity (APR: 1.3, 95% CI: 1.1–1.6, p<0.001) also had a higher prevalence of hypertension compared to people without alcohol drinking or sufficient physical activity. S2 and S3 Tables show the risk factors by sex.

## Discussion

Findings from this household survey reveal a high prevalence of NCD risk factors in a peri-urban and rural population in Eastern Uganda. We found that about two-thirds of the respondents had higher than normal BP (i.e., either prehypertension or hypertension). Most respondents were not taking an adequate amount of fruits/vegetables and over one-third were either overweight or obese. We also identified several risk factors for hypertension, including older age, alcohol drinking, low physical activity, overweight, and obesity. This study adds to the

**Table 4. Crude and adjusted prevalence ratios for hypertension.**

| Variable | | Unadjusted | | Adjusted | |
|---|---|---|---|---|---|
| | | CPR | p-value | APR | p-value |
| Age (year, ref.: 18–29 years) | 30–44 | 3.5*** (2.4,4.9) | < .001 | 3.1*** (2.2,4.4) | < .001 |
| | 45–59 | 5.9*** (4.2,8.2) | < .001 | 5.2*** (3.7,7.3) | < .001 |
| | 60+ | 10.1*** (7.3,13.9) | < .001 | 8.9*** (6.4,12.5) | < .001 |
| Male gender (ref.: Female) | | 0.9 (0.8,1.1) | .44 | | |
| Education (ref.: No education) | Primary | 0.6*** (0.5,0.7) | < .001 | 1.0 (0.9,1.2) | .89 |
| | Secondary | 0.3*** (0.3,0.4) | < .001 | 1.0 (0.8,1.2) | .72 |
| | Above Secondary | 0.5*** (0.4,0.7) | < .001 | 1.0 (0.7,1.4) | .95 |
| Location, Peri-urban (ref.: Rural) | | 0.9 (0.8,1.1) | .45 | | |
| Current tobacco smoker (ref = No) | | 2.0*** (1.5,2.6) | < .001 | | |
| Daily tobacco smoker (ref = No) | | 1.6* (1.1,2.3) | .024 | | |
| Current smokeless tobacco user (ref = No) | | 2.1* (1.1,4.0) | .025 | | |
| Daily smokeless tobacco user (ref = No) | | 0.3 (0.1,1.4) | .13 | | |
| Current tobacco (any) user (ref = No) | | 1.9*** (1.5,2.5) | < .001 | 1.2 (0.9,1.6) | .16 |
| Daily tobacco (any) user (ref = No) | | 1.4 (1.0,2.1) | .064 | | |
| Drank alcohol in past month (ref = No) | | 2.1*** (1.7,2.5) | < .001 | 1.6*** (1.3,2.0) | < .001 |
| Heavy episodic drinking* (ref = No) | | 2.0*** (1.6,2.5) | < .001 | | |
| Low fruit & vegetable consumption (ref = No) | | 1.1 (0.6,1.9) | .86 | | . |
| Add salt while eating (ref: Never/rarely) | Sometimes | 0.8* (0.7,1.0) | .024 | 1.0 (0.9,1.2) | .68 |
| | Often/always | 1.3 (0.9,1.8) | .2 | 1.6* (1.1,2.2) | .016 |
| Add salt while cooking | Sometimes | 0.7*** (0.6,0.9) | < .001 | | |
| | Often/always | 0.7** (0.6,0.9) | .002 | | |
| Eat processed foods high in salt | Sometimes | 0.8** (0.7,0.9) | .004 | | |
| | Often/always | 0.7 (0.5,1.0) | .068 | | |
| Poor physical activity (ref = No) | | 1.8*** (1.5,2.2) | < .001 | 1.3** (1.1,1.6) | .008 |
| Body mass index (ref: <25.0 kg/m2) | 25.0–29.9 | 1.4*** (1.2,1.7) | < .001 | 1.3** (1.1,1.5) | .006 |
| | 30 or more | 2.7*** (2.2,3.3) | < .001 | 2.0*** (1.6,2.4) | < .001 |

Abbreviations: CPR, crude prevalence ratio; APR, adjusted prevalence ratio.

growing body of literature investigating the burden of hypertension and NCD risk factors in LMICs.

The low fruit and vegetable consumption reported by our study was similar to the 2014 Ugandan STEPS NCD survey. The survey reported that nearly 88% of the participants had low fruit and/or vegetable consumption (defined as <5 servings on average per day) [10]. S4 and S5 Tables present a side-by-side comparison of 2014 national survey estimates with IMHDSS estimates for NCD risk factors in males and females, respectively, by age. Similarly, a recent cross-sectional national survey in Uganda similarly found that among participants from the Eastern region, the same region as Iganga-Mayuge, only 7.5% consumed ≥ 5 servings of fruits or vegetables per day [22]. Our study reveals a lower consumption of fruit and vegetables, which may be attributed to the dietary changes associated with increasing urbanization of rural and peri-urban areas, and perhaps a regional focus on producing commercial agricultural produce–for examples, cereals and roots–as opposed to fruits and vegetables [23]. A diet low on fruits and vegetables, but high on energy-rich foods are expected to be associated with higher levels of adiposity, body weight, and blood pressure [24, 25].

Another important finding was the high prevalence of hypertension within the Iganga Mayuge community, with nearly 17% and 50% of those participants having hypertension and

pre-hypertension, respectively. Though the prevalence of hypertension was lower than the 2014 Ugandan STEPS NCD, which reported that 26.5% of adults aged 18 to 64 years had it [10], in a peri-urban environment of Iganga-Mayuge the onset of hypertension, a cardiometabolic risk factor may observed over longer time periods following increases in body fat [25]. The high percentage of pre-hypertension in this community is worth noting, especially as some studies estimate that nearly 1 in 3 people with pre-hypertension develop hypertension within 4 years [26]. In our study, only 9.4% of survey participants had ever been diagnosed with hypertension, and nearly 60% had never had their BP measured. Low levels of awareness about adults' own hypertension status have been documented in Uganda [9, 24], including in the 2014 Ugandan STEPS survey which reported that only about 8% of participants were aware of their hypertension status [10], and in other Sub-Saharan countries [27, 28].

The low levels of awareness of one's hypertension status may reflect the health system's limited capacity to provide timely diagnostic, preventive, and treatment services to the general population [29]. Even within our study, we frequently had to refer participants who had been diagnosed with hypertension to the nearest district hospital because primary and community health centers often lacked diagnostic devices or were experiencing stockouts for hypertension treatment. To improve the availability of services in health facilities, the Ugandan Ministry of Health and NCD partners are already taking steps to improve population-level awareness around NCDs by updating clinical guidelines for managing major NCD conditions as well as developing and implementing programs to increase population-level awareness about NCDs and their risk factors [30]. While these activities demonstrate progress, deep-seated concerns remain among both government and external actors around the government's commitment to increase funding for NCD prevention and control in Uganda [7, 30].

Our study has several strengths. To our knowledge, this is the first study in Uganda since the 2014 Ugandan STEPS survey that systematically measured the major NCD risk factors in any region. In addition, a large population-based sample allows us to examine risk factors for NCDs in a peri-urban region in Uganda. Thirdly, our team of well-trained and experienced data collectors was critical in minimizing errors associated with taking anthropometry and BP measurements. Finally, our study aimed to reduce non-response rates by visiting households at least three times and at varying time points to reach the target survey participant.

Despite these advantages, our study is not without limitations. Though we can examine associations among different factors affecting NCDs, the cross-sectional design restricts us from making any causal inferences. Secondly, there may be potential for recall bias as many risk factors are self-reported by the participant. We further recognize that participants may have had difficulty in both assessing and recalling responses to certain questions, for example, around the actual number of servings of fruits and vegetables they consume on average in a day. A validation study may have addressed this concern; however, we tried to minimize this effect by using a survey that was based on a validated questionnaire. Thirdly, we recognize the potential misclassification of BP; however, we safeguarded against this by taking three BP measurements. Moreover, measuring BP in home setting instead of clinical setting can show different estimates, and standard guidelines recommend measuring BP in clinical settings for multiple times. However, the BP was measured in home setting for convenience [18, 19].

## Conclusion

Our study found a high prevalence of NCD risk factors in a peri-urban and rural population in Eastern Uganda, including indicators related to tobacco, diet, alcohol, and physical activity. We observed a high prevalence of pre-hypertension/hypertension highlighting the need to implement and scale-up population-level strategies to increase awareness about leading NCD

risk factors, especially the indicators studied here. A concomitant increase in health system's capacity to increase routine screening for hypertension and the provision of timely treatment will be critical for managing NCDs.

## Supporting information

**S1 Checklist. Inclusivity in global research.**
(DOCX)

**S1 Table. Demographic characteristics of study participants; Iganga-Mayuge, Uganda (Nov 2017-June 18).**
(DOCX)

**S2 Table. Crude and adjusted risk ratios for hypertension in females.**
(DOCX)

**S3 Table. Crude and adjusted risk ratios for hypertension in males.**
(DOCX)

**S4 Table. Comparison of 2014 national survey estimates with IM-HDSS estimates for NCD risk factors in males by age.**
(DOCX)

**S5 Table. Comparison of 2014 national survey estimates with IM-HDSS estimates for NCD risk factors in females by age.**
(DOCX)

## Acknowledgments

We would like to thank all data collectors, administrators and staff at Iganga-Mayuge Health and Demographic Surveillance Site for their support with the planning and implementation of the data collection activities.

## Author Contributions

**Conceptualization:** Dustin G. Gibson.

**Data curation:** Tryphena Nareeba, Collins Gyezaho.

**Formal analysis:** Dustin G. Gibson, Gulam Muhammed Al Kibria.

**Funding acquisition:** Dustin G. Gibson, Adnan Ali Hyder.

**Investigation:** Charles Ssemagabo, Judith Kaija Nanyonga, Elizeus Rutebemberwa.

**Methodology:** Dustin G. Gibson, Gulam Muhammed Al Kibria, Dan Kajungu, Elizeus Rutebemberwa.

**Project administration:** Dan Kajungu.

**Resources:** Tryphena Nareeba, Edward Galiwango.

**Supervision:** Dustin G. Gibson.

**Writing – original draft:** Dustin G. Gibson, Ankita Meghani.

**Writing – review & editing:** Charles Ssemagabo, Adaeze Wosu, Gulam Muhammed Al Kibria, George W. Pariyo, Dan Kajungu, Elizeus Rutebemberwa, Adnan Ali Hyder.

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
