## [Decision Letter · Decision Letter 0]

15 Feb 2024

PGPH-D-23-02022

The Epidemiology of Noncommunicable Disease Behavioral Risk Factors and Hypertension: A Cross-sectional Study from Eastern Uganda

Dear Dr. Kibria,

Thank you for submitting your manuscript to PLOS Global Public Health. After careful consideration, we feel that it has merit but does not fully meet PLOS Global Public Health’s publication criteria as it currently stands. Therefore, we invite you to submit a revised version of the manuscript that addresses the points raised during the review process.

We look forward to receiving your revised manuscript.

Kind regards,

Dickson Abanimi Amugsi, PhD

Academic Editor

Journal Requirements:

Additional Editor Comments (if provided):

Reviewers' comments:

Reviewer's Responses to Questions

**Comments to the Author**

1. Does this manuscript meet PLOS Global Public Health’s publication criteria? Is the manuscript technically sound, and do the data support the conclusions? The manuscript must describe methodologically and ethically rigorous research with conclusions that are appropriately drawn based on the data presented.

Reviewer #1: Yes

Reviewer #2: Yes

Reviewer #3: Yes

2. Has the statistical analysis been performed appropriately and rigorously?

Reviewer #1: Yes

Reviewer #2: Yes

Reviewer #3: Yes

3. Have the authors made all data underlying the findings in their manuscript fully available (please refer to the Data Availability Statement at the start of the manuscript PDF file)?

Reviewer #1: Yes

Reviewer #2: Yes

Reviewer #3: Yes

4. Is the manuscript presented in an intelligible fashion and written in standard English?

Reviewer #1: Yes

Reviewer #2: Yes

Reviewer #3: Yes

5. Review Comments to the Author

Reviewer #1: The manuscript seems to be technically sound, and the data supports the conclusions. The methods used can be clearly understood and the study can be replicated. Appropriate ethical considerations have been taken care off. Conclusions are appropriate and drawn based on the data presented. Refer to the attached manuscript for recommendations.

Reviewer #2: Abstract

Consider revising your abstract to be reader friendly by indicating and maybe reducing font size of statistical indicators and figures .

Introduction

In the problem statement , authors raised the issue of suboptimal NCD risk factors surveillance efforts and in their methods said the study was nested in a longitudinal cohort( surveillance ) data . So which gap is the paper addressing since surveillance data was available ?

Reviewer #3: A very good study and timely,

A few comments to note, the authors have indicated that the BP was measured three times but did not indicate which arm was used to measure the BP, Art is known that BP differs in the two arms. How can the reader confirm that BP was measured consistently in the same arm for all the respondents?

Why were pregnant women excluded yet they have more challenges with BP? And why were they part of the sample if BP and weight were not going to be measured?

it will be good to capture in the literature the pros and cons of home BP measurement vs health facility.

It will be good to understand how a serving was defined to allow the respondent to recall close to accurately. How was serving of vegetables designed given the challenges of dietary recalls.

6. PLOS authors have the option to publish the peer review history of their article (what does this mean?). If published, this will include your full peer review and any attached files.

**Do you want your identity to be public for this peer review?** For information about this choice, including consent withdrawal, please see our Privacy Policy.

Reviewer #1: **Yes: **Abdul Nazer Ali

Reviewer #2: No

Reviewer #3: No

---

## [Decision Letter · Decision Letter 1]

2 Apr 2024

The Epidemiology of Behavioral Risk Factors for Noncommunicable Disease and Hypertension: A Cross-sectional Study from Eastern Uganda

PGPH-D-23-02022R1

Dear Kibria,

We are pleased to inform you that your manuscript 'The Epidemiology of Behavioral Risk Factors for Noncommunicable Disease and Hypertension: A Cross-sectional Study from Eastern Uganda' has been provisionally accepted for publication in PLOS Global Public Health.

Best regards,

Dickson Abanimi Amugsi, PhD

Academic Editor

Reviewer Comments (if any, and for reference):

Reviewer's Responses to Questions

**Comments to the Author**

1. If the authors have adequately addressed your comments raised in a previous round of review and you feel that this manuscript is now acceptable for publication, you may indicate that here to bypass the “Comments to the Author” section, enter your conflict of interest statement in the “Confidential to Editor” section, and submit your "Accept" recommendation.

Reviewer #1: All comments have been addressed

Reviewer #3: All comments have been addressed

2. Does this manuscript meet PLOS Global Public Health’s publication criteria? Is the manuscript technically sound, and do the data support the conclusions? The manuscript must describe methodologically and ethically rigorous research with conclusions that are appropriately drawn based on the data presented.

Reviewer #1: Yes

Reviewer #3: Yes

3. Has the statistical analysis been performed appropriately and rigorously?

Reviewer #1: Yes

Reviewer #3: Yes

4. Have the authors made all data underlying the findings in their manuscript fully available (please refer to the Data Availability Statement at the start of the manuscript PDF file)?

Reviewer #1: Yes

Reviewer #3: Yes

5. Is the manuscript presented in an intelligible fashion and written in standard English?

Reviewer #1: Yes

Reviewer #3: Yes

6. Review Comments to the Author

Reviewer #1: (No Response)

Reviewer #3: regarding BP measurement at home, it can also be indicated that home measurements reduce white coat BP ( they seek references that talk of white coat BP. But also home measurement empowers the Hypertensive patients to take control of their health

7. PLOS authors have the option to publish the peer review history of their article (what does this mean?). If published, this will include your full peer review and any attached files.

**Do you want your identity to be public for this peer review?** For information about this choice, including consent withdrawal, please see our Privacy Policy.

Reviewer #1: **Yes: **Abdul Nazer Ali

Reviewer #3: No
